# Shock-Impacts and Vibrational *g*-Forces Can Dislodge *Bacillus* spp. Spores from Spacecraft Surfaces

**DOI:** 10.3390/microorganisms11102421

**Published:** 2023-09-28

**Authors:** Andrew C. Schuerger, Adriana V. Borrell

**Affiliations:** Department of Plant Pathology, University of Florida, Space Life Sciences Lab, 505 Odyssey Way, Exploration Park, Merritt Island, FL 32953, USA

**Keywords:** planetary protection, astrobiology, spacecraft microbiology, mars, special regions

## Abstract

Mars spacecraft encounter numerous *g*-loads that occur along the launch or landing vectors (called axial vectors) or along lateral off-axes vectors. The goal of this research was to determine if there was a threshold for dislodging spores under brute-force dynamic shock compressional impacts (i.e., henceforth called shock-impacts) or long-term vibrationally induced *g*-loads that might simulate spacecraft launches or landings profiles. Results indicated that spores of *Bacillus subtilis* 168 and *B. atrophaeus* ATCC 9372 were dislodged from ChemFilm-coated aluminum coupons during shock impact events of 60 *g*’s or higher. In contrast, the threshold for dislodging *B. pumilus* SAFR-032 spores was approx. 80 *g*’s. Vibrational *g*-loading was conducted at approx. 12–15 *g*’s (z-axis) and 77 Hz. All three *Bacillus* spp. exhibited very modest spore dislodgement at 1, 4, or 8 min of induced vibrational *g*-loads. However, the numbers of spores released depended on the Earth’s *g*-vector relative to the bacterial monolayers. When the experimental hardware was placed in an ‘*Up*’ orientation (defined as the spores sat on the upper surface of the coupons and the coupons pointed up and away from Earth’s g-vector), zero to only a few spores were dislodged. When the experimental hardware was inverted and the coupon surfaces were in a ‘*Down*’ orientation, the number of spores released increased by 20–30 times. Overall, the results of both assays suggest that spores on spacecraft surfaces will not likely be dislodged during nominal launch and landing scenarios, with the exception of jettisoned hardware (e.g., heat shields or backshells) during landing that might hit the Martian terrain at high *g*’s. However, off-nominal landings hitting the Martian surface at >60 *g*’s are likely to release low numbers of spores into the atmosphere and regolith.

## 1. Introduction

The Mars spacecraft Pathfinder, Spirit, Opportunity, Phoenix, Curiosity, and InSight were rated as either Category IVa or IVc (Phoenix) missions, and as such, their bioburdens were limited to ≤5 × 10^5^ total spores per vehicle or ≤300 spores/m^2^ at launch [1]. Published data indicate that the bioburdens estimated at launch for these spacecraft met or exceeded the guidelines (i.e., lower bioburdens than required [2,3]). However, spacecraft bioburden models [4,5,6] suggest that total spacecraft bioburdens of all microbiota are likely to be up to three orders of magnitude higher than the planetary protection guidelines required for spore-forming species.

After launch, Mars spacecraft spend 6–8 months in a cruise phase in which space biocidal factors begin to inactivate bioburdens on external and internal surfaces [6,7]. It is likely that 1–3 orders of magnitude reductions in viable bioburdens can be achieved during the Earth–Mars cruise phase [7]. However, this would leave many individual viable cells, spores, and cell–spore aggregates to persist through the entry, descent, and landing (EDL) profiles of spacecraft.

Spacecraft EDL scenarios are typically divided into landings with air-bag delivery systems (e.g., Pathfinder, Spirit, Opportunity rovers) or landings with active descent engine braking at the surface (e.g., Phoenix, Curiosity, InSight, Perseverance spacecraft). The air-bag delivery systems create high-*g* spikes as the spacecraft bounces along the landing ellipse. For example, the Pathfinder rover experienced 14 bounces, with the first bounce creating a 15.5 *g* dynamic shock compressional impact (henceforth called shock-impacts) to the landed systems [8]. All subsequent bounces created a diminishing series of loads that averaged approx. 8–10 *g*’s. In contrast, active descent landing configurations may have a high-*g* phase during landing, but it is generally observed during the hypersonic phase just prior to parachute deployment. For example, the Phoenix lander experienced a slow build-up of *g*-loads to 9.2 *g*’s during its hypersonic phase [9]. However, in the off-nominal landings of spacecraft (e.g., Beagle 2, Mars Polar Lander) and in the nominal impacts of jettisoned subsystems like the Perseverance aeroshell (Figure 1) or Opportunity heat shield (Figure 2), the *g*-forces can be high enough to cause the disintegration of spacecraft components.

The goals of the current study were to determine if *g*-force thresholds exist for dislodging *Bacillus* spp. spores from aluminum coupons under shock impact or vibrational *g*-loading conditions. Vibrational *g*-loads can be encountered during the Earth launch or EDL of planetary spacecraft. If shock-impacts or vibrational-*g* loads experienced by spacecraft during launch or if EDL cause the release of microbiota from surfaces, the dislodged cells/spores may have consequences for predicting the forward contamination risks to Special Regions [10] on Mars.

The nomenclature of Hazel et al. [11] is adopted here for differentiating *static compressional g-forces* (i.e., high pressures imposed on microbiota through devices that create and then hold the samples at elevated pressures (not tested here)) and *dynamic shock compressional g-forces* (i.e., very-short-duration high-*g* events). We further simplify the second term to ‘shock-impacts’ for the discussions below. Lastly, we use the term ‘*threshold*’ here in its generic form defined as ‘…a level, point, or value above which something is true or will take place…’ (Merriam-Webster Dictionary (online at https://www.merriam-webster.com/dictionary/threshold (accessed on 17 September 2023)).

## 2. Methods

### 2.1. Microbial Protocols

Endospores (henceforth spores) of three *Bacillus* spp. were tested for their ability to resist removal from aluminum coupons by either shock impact or vibrational *g*-loading. Spores of *B. subtilis* 168 and *B. pumilus* SAFR-032 were produced according to the protocols described by Schuerger [12] and Schuerger and Headrick [13]. Spores of *B. atrophaeus* ATCC 9372 were purchased from Mesa Labs (#SSGE/7, Lakewood, CO, USA).

All spores were stored at 4 °C until used. Spores were diluted in sterile deionized water (SDIW) and calibrated with a UV–VIS spectrometer at 600 nm until the desired spore densities were achieved (~2 × 10^7^ spores/mL). Subsequently, 100 µL of spore suspensions from each *Bacillus* spp. was pipetted onto individual aluminum coupons, forming 1.2-cm diameter droplets with ~2 × 10^6^ spores/coupon, stored in closed Petri dishes overnight at 22–24 °C (i.e., to allow spores to settle to the aluminum surfaces of the coupons), and then air-dried in a biosafety hood (model 440–600, NuAire Class II, Type A2 hood, Plymouth, MN, USA) for 3–4 h the next day. Bacterial spores—when dried—formed uniform monolayers on the ChemFilm-coated aluminum coupons (see [12,13] for monolayer protocols and SEM images).

Aluminum coupons were treated with a Class 1A ChemFilm coating (i.e., a chromium oxide coating; syn. Iridite), sterilized at 130 °C dry heat for 48 h, and used as the substrate for holding spores. Coupons measured 54 × 17 × 0.5 mm. ChemFilm coatings on aluminum coupons were applied by Synergy Metal Finishing, Inc. (Titusville, FL, USA).

Individual Microbial Sample Holders (MSH; Figure 3 (described by [14])) were fitted with three doped coupons of specific *Bacillus* spp. All MSH units were dry heat sterilized at 130 °C for 2–3 d prior to use. The coupons were randomized in the MSH holders and screwed down to the baseplates with stainless steel screws. Once the coupons were secured to the MSH baseplates, the tops were secured with stainless steel screws. In the tests described below, no coupons, MSH lids, or screws became loose during the high *g*-force assays. The assembly of the MSH units was completed under aseptic conditions within the NuAire biosafety hood.

### 2.2. Shock Impact g-Forces Experiments

Shock-impact events refer to tests in which assembled MSH units containing doped coupons were accelerated to high speeds and targeted to impact a stationary stainless steel plate (measuring 30 × 30 × 1.3 cm), coming to an abrupt stop on the upper surface of the steel plate. The hardware for the shock impact assays is depicted in Figure 3A. Three-axes *g*-loads for a single shock impact assay are given in Figure 4 as an example of a typical test. Different impact speeds induced divergent shock-impact *g*-loads.

First, the assembled MSH units were inverted such that the spores on coupon surfaces faced down. Second, three-axes accelerometers (X200-4, Gulf Coast Data Concepts, Waveland, MS, USA) were attached to the bottoms of individual MSH units (now facing upward). The X200-4 accelerometers (as depicted in Figure 3A,B) were able to measure instantaneous *g*-loads at 400 Hz in three axes (i.e., x-axis *g*’s indicated front-to-back accelerations, y-axis *g*’s indicated left-to-right accelerations, and z-axis *g*’s indicated up-to-down accelerations). Each axis could record *g*-loads up to 200 *g*’s. All programming and calibration of the X200-4 accelerometers were conducted as per the factory manuals. Third, the MSH units—plus X200-4 accelerometers—were placed on the steel plate. The assemblies were raised to various elevations up to 1 m and then manually accelerated downward to impact the upper surface of a stainless-steel plate. The approach would test the removal of spores downward and away from the aluminum coupons; thus, it was assumed that dislodged spores would adhere to the inner MSH lid surfaces. Individual MSH units with doped coupons were exposed to only one shock impact event per assay. Between 20 and 21 independent assays (i.e., replicates) were conducted for each *Bacillus* spp. (see Appendix A) and plotted together as scatter plots for each species (see below).

Once MSH units had received various shock-impacts on the steel plate, they were returned to the NuAire biosafety hood and aseptically assayed in the following manner (Figure 3C,D). First, the MSH units were placed in the normal ‘*lids up*’ configuration. The screws holding the lids to the baseplates were removed, the lids were inverted, and sterile polyester swabs were used to swab the inner surfaces of the MSH lids. The polyester swabs (#25-806-1PD, Puritan Medical Products Co., LLC., Guilford, ME, USA) were wetted by dipping in filter-sterilized (0.2 µm) 1x phosphate buffered saline (PBS; pH = 7.2). Immediately after swabbing the MSH inner lid surfaces, the polyester-tipped applicators would be used to streak the upper surfaces of two tryptic soy agar (TSA) Petri dishes. All cultures were incubated at 30 °C for 48 h and evaluated for germination and growth of *Bacillus* spp. spores.

The polyester swab protocol was designed as a semi-quantitative assay in which dislodged spores could be detected inside the MSH units. However, the assay was likely not able to collect 100% of the dislodged spores. In preliminary trials, increasing the numbers of TSA plates did not appreciably increase the numbers of recovered spores. We believe that using one polyester swab and two TSA plates recovered approx. 80% of the total numbers of dislodged spores per MSH unit per assay.

### 2.3. Vibrational g-Force Assays

The shock impact assays measured the effects of instantaneous high *g*’s in the z-axis on the dislodgement of *Bacillus* spp. spores from ChemFilm-coated aluminum coupons. A separate assay was developed to measure long-term ‘*vibrationally induced*’ *g*’s on the removal of spores from ChemFilm-coated coupons. The long-term vibrational *g*’s were created by placing assembled MSH units (i.e., with doped coupons) on a specially designed aluminum mounting plate secured to a mechanical dry-sieving apparatus (Figure 3B; model AS200, Retsch, Haan, Germany). The mounting plate had multiple locations for mounting up to three MSH units. However, in preliminary tests, a single center-mounted MSH unit was found to give the most uniform and controllable long-term vibrational *g*-loads. In preliminary assays, settings on the mechanical sieving apparatus were determined that would yield approx. 10–12 *g*’s in the z-axis direction (i.e., away from the coupons).

Two separate orientations of the MSH units were tested for the vibrational assays. First, the MSH units were mounted on the vibration table with the spore monolayers pointing either upward (i.e., *Up* configuration: MSH unit mounted with lids pointing upward) or downward (i.e., *Down* configuration: MSH unit mounted with lids pointing downward). Thus, the effects of long-term vibrational *g* loading on spore removal away from coupons could be measured when Earth’s 1-*g* environment was part of the assays.

Negative controls for both *g*-force assays were completed by preparing and moving assembled MSH units to either the steel plate (shock-impact *g*-force assays) or Retsch sieving apparatus (vibrational *g*-loads), but in which no *g*-forces were applied. In all cases, no spores were detected in the control assays that would indicate that the spores were dislodged from normal handling of the MSH + coupon assemblies.

Each MSH unit contained three coupons doped with individual *Bacillus* sp. (i.e., each species was run separately to avoid cross-contamination of coupons). Subsequently, each MSH and *Bacillus* sp. combination was conducted twice in both vibrational configurations to yield a total of six replicates per treatment (n = 6). Examples of the vibrational *g*-loads for all three axes are given for a typical test in Figure 5.

### 2.4. Statistical Analysis of the g-Force Assays

Data for the shock-impact *g*-force assays were rendered as scatter plots (Figure 6). The goal was to identify the thresholds for the three *Bacillus* spp. above which spores would be dislodged from the ChemFilm Class 1A surfaces. No statistical analyses were conducted on the data in Figure 6.

In contrast, data from the vibrational *g*-loading assays (Figure 7) were analyzed with the PC-based Statistical Analysis System (SAS) version 9.4 (SAS Institute, Inc., Cary, NC, USA). Data in Figure 7 were treated with a 0.50-power transformation to induce homogeneity of treatment variances. Transformed data were analyzed with ANOVA and a protected least-squares mean separation test. Treatments for individual *Bacillus* spp. with divergent letters were significantly different at *p* ≤ 0.05 (n = 6). Data are presented as untransformed values.

## 3. Results

### 3.1. Accelerometer Data

Accelerometer data for the shock impact assays indicated that the primary *g*-loads occurred in the z-axis and that some lateral *g*-loading was observed throughout the assays (Figure 4). The durations of the events lasted approx. 50 ms per impact. In contrast, the *g*-loading for the vibrational assays indicated that MSH units were exposed to approx. 1.5 *g*’s, 3–4 *g*’s, and 12–14 *g*’s, respectively, for the x-, y-, and z-axes (Figure 5) at approx. 77 Hz (+/− 5 Hz; z-axis). The peak *g*-forces in all three axes initially peaked slightly above these values. However, within a few hundred milliseconds, the vibration table would stabilize in all three axes.

### 3.2. Shock impact Assays

Due to the nature of the shock impact assays (i.e., using manual acceleration of the MSH units down and towards the steel plate) it was difficult to achieve precisely the same *g*-forces among numerous events. Thus, an approach was taken in which individual shock impact events for individual assays were plotted for the *g*-loading (abscissa axis) and recovery of dislodged spores (ordinate axis). For *B. subtilis* and *B. atrophaeus*, spores were recovered in shock impact assays when the z-axis acceleration was ≥60 *g*’s (Figure 6). In contrast, the spores of *B. pumilus* did not begin to be dislodged until the z-axis *g*-loading was approx. 80 *g*’s (Figure 6B). There were general trends for all three *Bacillus* spp. to have more spores dislodged as *g*-loads increased above these thresholds. Assays were completed in which the z-axis loads approached the upper limit of the X200-4 accelerometers (i.e., 200 *g*’s).

### 3.3. Vibrational g-Force Assays

In longer-term vibrational *g*-force assays, the results were, in general, more complex. The vibrational assays were not designed to determine if there were thresholds for spore removal at specific times versus *g*-loading combinations. Instead, all vibrational assays were conducted with an upper limit of 15 *g*’s for all three axes over the course of 8 min. As discussed above, the max *g*-loads for the vibrational assays occurred for the z-axis at approx. 12–14 *g*’s (stabilized vibrations).

For all three *Bacillus* spp., spore removal was either zero or extremely low when the spore monolayers were in the *Up* orientation (Figure 7). In the *Up* orientation, spores would have had to become dislodged from the aluminum coupons and then be transported away from coupons and adhere to the inside surfaces of the MSH lids. In contrast, spores were detected in greater abundance when the MSH units were exposed to vibrational *g*-loads when in the *Down* orientation for all three timesteps. The added 1-*g* vector of Earth’s gravitational field was able to impart the extra force necessary to allow dislodged spores to traverse the approx. 7 mm distance from the coupon surfaces to the inner sides of the MSH lids. It is plausible that spores were dislodged in the *Up* orientation, but the spores were unable to move against Earth’s 1-*g* field in the vibrational assays.

In general, the numbers of dislodged spores increased with the increasing time exposed to the vibrational *g*-loads for all three *Bacillus* spp. (Figure 7). The highest spore removal rates were observed for monolayers exposed to 8 min for *B. subtilis* and *B. atrophaeus*, with an excess of 30 spores dislodged per MSH unit (Figure 7A,C). However, the maximum number of spores for *B. pumilus* in any of the timesteps was approx. 10 spores dislodged for either 4- or 8-min assays (Figure 7B). Results from both assays suggested that the spores of *B. pumilus* SAFR-032 attached to the Class 1A ChemFilm-coated aluminum coupons adhered to coupon surfaces slightly more effectively than either *B. subtilis* 168 or *B. atrophaeus* 9372. Asterisks in Figure 7 indicate treatments in which no viable spores were recovered on TSA plates in the vibrational *g*-loading assays; thus, no spores were assumed to have been dislodged.

## 4. Discussion

Modeling the forward contamination risk of the Martian terrain requires a variety of factors to be defined and characterized. Typical predictions rely almost exclusively on characterizing the launched bioburdens on spacecraft surfaces prior to launch [1]. However, if >99.99% of the bioburdens on Mars rovers and landers remain on the spacecraft surfaces, then the forward contamination of the local terrain is likely to be extremely low. Estimates on how much bioburden can be dislodged from spacecraft during Earth-launch and Mars EDL profiles are limited, and dislodged spores may be a dominant factor in predicting the risks to Special Regions on Mars. One example is a study in which descent engine exhaust was shown to scour landing struts on the Phoenix lander [15]; thus, it would have removed and dispersed many viable cells/spores remaining on external surfaces at landing.

The experiments described here report for the first time that high *g*-loading on spacecraft can dislodge bacterial spores from analog spacecraft surfaces. The research considered two possible *g*-loading scenarios: high-*g* shock-impacts and vibrational *g*-loading. High-*g* shock-impact events would be applicable for the off-nominal impacts of spacecraft or the expected impacts of jettisoned subsystems (e.g., backshells or heat shields of rovers). Vibrational *g*-loading would occur in both Earth-launch and Mars EDL profiles.

For the bacteria tested here, 60-*g* thresholds were observed for *B. subtilis* 168 and *B. atrophaeus* 9372 spores on ChemFilm-coated aluminum coupons during shock-impact events. The high-*g* threshold for dislodging spores was slightly higher at 80 *g*’s for *B. pumilus* SAFR-032. However, what was remarkable was the observation that the numbers of dislodged spores ranged from single digits to 10s or low-100s of spores for each impact event, even up to 190 *g*’s for the three *Bacillus* spp. tested (Figure 6). Recall that the population densities of spores on coupons were ~2 × 10^6^ at T = 0. Thus, the vast majority of spores remained firmly attached to ChemFilm-coated surfaces with all three *Bacillus* spp. as shock-impact forces were increased up to 190 *g*’s.

In contrast, continuous vibrational *g*-loading of the same *Bacillus* spp. and hardware produced dislodged spores at lower *g*-forces (z-axis) of up to ~12–14 *g*’s if the vibrations were extended for 1 to 8 min and the MSH units were inverted on the vibration table (Figure 7). Spores dislodged during the three-axes vibrations (inverted MSH units) increased with longer timesteps for all three *Bacillus* spp. tested. Surprisingly, very few to zero spores of all three *Bacillus* spp. were dislodged if the ChemFilm-coated coupons were in the upright configuration. The results suggest that vibrational *g*-loading is more effective in dislodging spores when the surfaces are concomitantly inverted relative to Earth’s planetary *g*-vector.

Most launch vehicles incur three-axes vibrations for approx. 8 min during launch to low-Earth orbit (LEO). However, only a few spores per MSH unit were reported at the upper range of time under vibrational *g*-loading. For example, approx. 30 and 35 spores were dislodged for *B. subtilis* 168 and *B. atrophaeus* 9372, respectively, per coupon on samples exposed to three-axes vibrations for 8 min (i.e., *Down* MSH units) (Figure 7A,C). In contrast, only approx. 10 spores per MSH unit were dislodged for *B. pumilus* SAFR-032 when the coupons were exposed to vibrations at 4 and 8 min (i.e., *Down* MSH units) (Figure 7B). These results suggest that spores of *B. pumilus* adhere more strongly to Class 1A ChemFilm aluminum compared to the other two species, which is consistent with earlier research on spore attachment to aluminum surfaces [12].

The three key assumptions in these experiments were as follows: (1) bioburdens on Mars spacecraft are attached as low-density single cells/spores or small aggregates of the same [16]; (2) that ChemFilm-coated aluminum coupons were good proxies for the average material of a Mars spacecraft [13,17]; and (3) the *g*-loads in these experiments were representative of both the launch and EDL profiles of Mars spacecraft (see below). Furthermore, it is unlikely that the shock-impacts of vibrational *g*-loads tested here directly impacted microbial survival because numerous studies have shown reasonable rates of microbial survival up to 57–78 GPa [18,19]; i.e., significantly higher *g*-loads for shock impact events than tested here.

Furthermore, ChemFilm-coated (Class 1A) aluminum coupons were used here as the test material for both *g*-loading assays. ChemFilm was selected based on previous work with multiple analog spacecraft components as support materials for microbial assays [13,17]. However, it is plausible that spore adhesion might diverge from these results (e.g., Figure 6 and Figure 7) if other spacecraft materials or microbial species are tested. For example, four forces may be involved in the attachment of particles (including microbes) to aluminum and include contact potential, electrostatic, van der Waals, and water capillary forces [20]. In addition, secreted cell/spore mucilage might contribute to adhesion to spacecraft materials. It is likely that diverse spacecraft components and materials will vary widely with respect to these properties, which would alter how particles—including bacteria—might adhere to surfaces.

New experiments that examine other spacecraft materials and microbial species are required to fully characterize how launch and EDL *g*-loading can affect spore removal from planetary spacecraft. The results presented herein should be viewed as preliminary data on how terrestrial microorganisms can adhere to planetary spacecraft during prelaunch processing and how the same bioburdens might be dislodged during launch and EDL events.

Lastly, if the data in Figure 6 and Figure 7 were replotted using the common approach of representing the removal of particles from surfaces as fractional terms (i.e., called ‘*particle removal fractions*’) [21,22], the results would indicate that the particle removal fractions would be between 10^−4^ (low 100s of recovered spores) and 10^−6^ (low single digits). These values are extremely low compared to assays that seek to identify a ‘*threshold velocity* or *shear stress*’, in which 50% of particles are removed at a given velocity in a shearing fluid [22]. Here we sought the thresholds (i.e., generic definition) at which single digits and 10s to low-100s of spores were initially dislodged from spacecraft surfaces. We call attention to these divergent definitions to avoid confusion.

### 4.1. Implications for Spacecraft during Launch

Possible effects of shock-impact and vibrational *g*-loads for launch are derived from data presented in the Delta II [23] and Atlas V [24] user guides. Pathfinder, Spirit, Opportunity, and Phoenix spacecraft were launched on Delta II rockets, and Insight and Perseverance were launched on Atlas V rockets. We assume here that the data given in these two user guides from United Launch Alliance (ULA) are averages of these missions.

First, spacecraft that weigh approx. 1500 kg in a Delta II launch configuration (i.e., the assumed average weight of the spacecraft listed above for the Delta II) will encounter a steady-state acceleration of approx. 7 *g*’s during launch at the peak acceleration of the spacecraft (i.e., at the end of first-stage main engine cutoff (MECO) [23]). However, the acceleration *g*-forces are not shock-impact *g*-loads; they build up slowly as the Delta II pushes through decreasing air resistance during launch. Furthermore, during Delta II launches, spacecraft typically encounter a variable vibrational *g*-loading environment up to 2.2 *g*’s (120 Hz) in the thrust vector and up to 0.85 *g* (120 Hz) in lateral vectors [23].

The launch performance of the Atlas V vehicle [24] is similar to that of the Delta II given above. The Atlas V experiences a very slow increase in acceleration up to 5 *g*’s for the thrust vector and up to 2 *g*’s for lateral vectors during launch, but the *g*’s are not shock-impact forces. In addition, the Atlas V payloads can experience low vibrational *g*-forces between 50 and 100 Hz that peak at approx. 1 *g* during launch.

The vibrational *g*-load frequencies tested here (12–14 *g*’s at 77 Hz; Figure 5) were in the same ranges of the Delta II and Atlas V vibrational frequencies but much more severe in the *g*-loads (i.e., 12–14 *g*’s here versus no more than 2.2 *g*’s in the thrust vector of a Delta II). First, our results suggest that *Bacillus* spores are unlikely to be dislodged from payload surfaces during launch on Delta II or Atlas V rockets. One exception to this conclusion might be unique staging events that might have unusually strong shock-impact forces with engine cutoffs and second- or third-stage booster ignitions. The model AS200 Retsch vibration table used here (Figure 3B) was very limited in its ability to adjust the *g*-forces in the assays. Thus, future vibrational *g*-loads at approx. 2.5 *g*’s and up to 120 Hz (i.e., similar to the Delta II and Atlas V launch vehicles) should be tested to determine if lower *g*-forces but higher frequencies result in the dislodgment of bacterial spores from spacecraft surfaces.

### 4.2. Implications for Mars Spacecraft during EDL

There are two basic architectures for landing spacecraft on Mars: air-bag bounce landings or active descent engine landings. The former can experience shock-impact forces during the bounces, but the latter approach is intended to gently land vehicles on the terrain.

The best data for the air-bag bounce EDL approach were found for the Pathfinder spacecraft [8]. The Pathfinder vehicle bounced 14 times on the Martian terrain before coming to rest. The initial bounce reached 15.5 *g*’s, the subsequent seven bounces peaked between 10 and 12 *g*’s, and the final six bounces were approx. 5 *g*’s. The shock impact *g*-force thresholds for dislodging spores described here were not reached on any of the Pathfinder bounces. Thus, it is unlikely that *Bacillus* spores that might have been present on internal spacecraft surfaces were dislodged during the Pathfinder EDL profile.

In addition, all spacecraft entering the Martian atmosphere slowly build-up *g*-loads along the entry vector during the hypersonic phase as atmospheric density increases. The peak *g*-loads were 16, 12.5, and 9.2 for the Mars spacecraft Pathfinder, Curiosity, and Phoenix, respectively [8,9,25]. However, these *g*-loads built up slowly and were not shock-impact events. Thus, for nominal EDL profiles, spores are unlikely to be dislodged from Mars spacecraft, even if air-bag-bounced landings are employed.

This leaves us to consider either off-nominal landings (e.g., Beagle 2 and Mars Polar Lander) or the impacts of jettisoned subsystems like the Perseverance backshell (Figure 1) or the Opportunity heat shield (Figure 2). In preliminary analysis of the impact *g*-loads for the Perseverance backshell, the structure likely hit the terrain at approx. 62 *g*’s (A. Chen, JPL-Cal-tech, personal communications), consistent with the thresholds—described here—for dislodging *B. subtilis* and *B. atrophaeus* spores from aluminum coupons (Figure 6). Examination of the image in Figure 1 clearly shows that the impact forces were adequate to crack and flatten the Perseverance backshell. We conclude here that the forces were also adequate to likely dislodge low numbers of spores from the structures. Furthermore, the Opportunity heat shield (Figure 2) broke up into two major components upon impact. Both examples of these expected impacts of jettisoned subsystems clearly show numerous pieces and parts strewn around the impact sites.

## 5. Conclusions

Mars spacecraft that successfully complete a nominal EDL profile are unlikely to experience *g*-loads that will dislodge spores into the Martian atmosphere or terrains. The exceptions to this rule might be jettisoned subsystems (e.g., backshells and heat shields described above) that are released above the Martian surface and have no active means of slowing down before impact, as well as landing struts and pads exposed to active descent engines. In the latter cases, bacterial spores are likely dislodged from spacecraft surfaces and may be immediately caught up in the swirling winds at the landing sites. In addition, bacterial spores could be released directly to the Martian regolith below the crashed components. Further research is suggested in order to characterize whether dislodged spores would survive long enough in the winds or regolith to encounter benign conditions and permit the growth and cellular replication of terrestrial microorganisms on Mars.

Furthermore, the tests presented here were done in Earth lab atmospheric conditions of 1013 mbar, 45% RH, room temperature between 22 and 24 °C, and with a gas composition of 78% to 21% for *p*N_2_ and *p*O_2_ gases. These conditions are not present on the Martian surface. Thus, additional research is suggested in order to repeat and extend these experiments into measuring the *g*-forces required to dislodge spores under simulated Martian conditions. The equipment to run these experiments under Martian atmospheric conditions was not available.

Lastly, future research is suggested in order to evaluate the adhesion and dislodgement of vegetative cells of bacteria and archaea—plus spores of eukaryotes like fungi—that might be present as bioburdens on spacecraft surfaces. Our initial results, presented here, suggest that some components hitting the Martian terrain could release spores if the shock-impact *g*-forces are greater than 60 *g*’s, but further study is warranted.

## Figures and Tables

**Figure 1 microorganisms-11-02421-f001:**
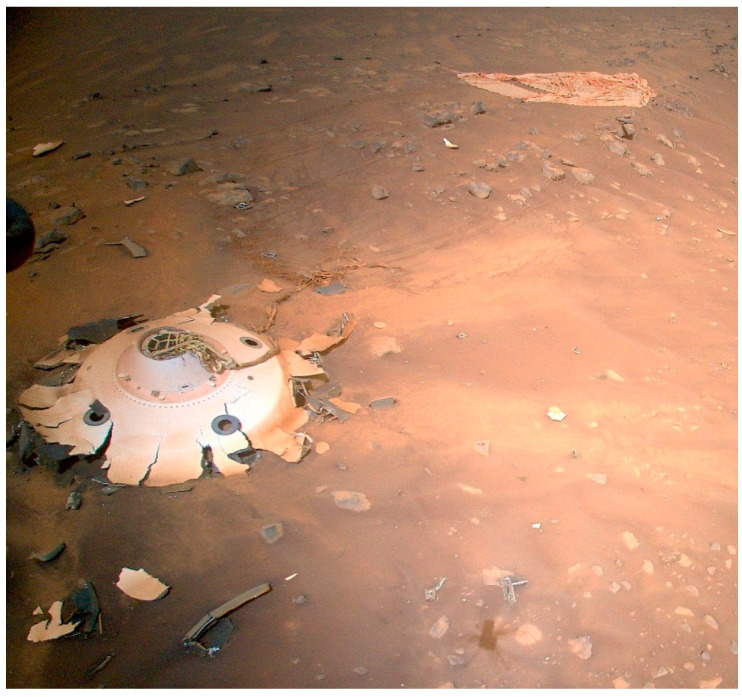
Mars surface debris field created by a 62-*g* impact of the Perseverance backshell during EDL on 18 February 2021. This image was taken by the Perseverance helicopter on sol 414 (20 April 2022) (photo credit: NASA/JPL-Caltech; image 2-PIA25217).

**Figure 2 microorganisms-11-02421-f002:**
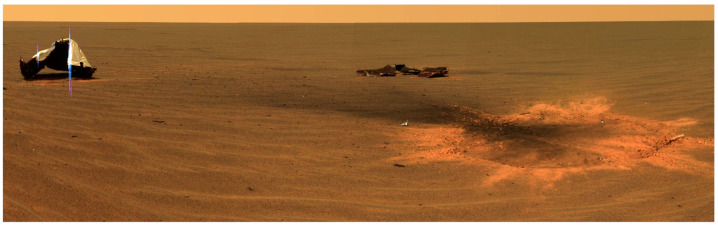
The Opportunity heat-shield impact site on Meridiani Planum on Mars. The heat shield impact site was imaged on sol 330 on 28 December 2004 by the rover and showed disintegration of the heat shield into two main components and a variety of smaller debris. However, the impact did not create a crater. The montage was created by JPL personnel and was found on the website https://www.jpl.nasa.gov/missions/mars-exploration-rover-opportunity-mer (accessed on 17 September 2023). The image was lightened slightly and cropped from the original downloaded file (photo credit: NASA/JPL-Caltech/Cornell).

**Figure 3 microorganisms-11-02421-f003:**
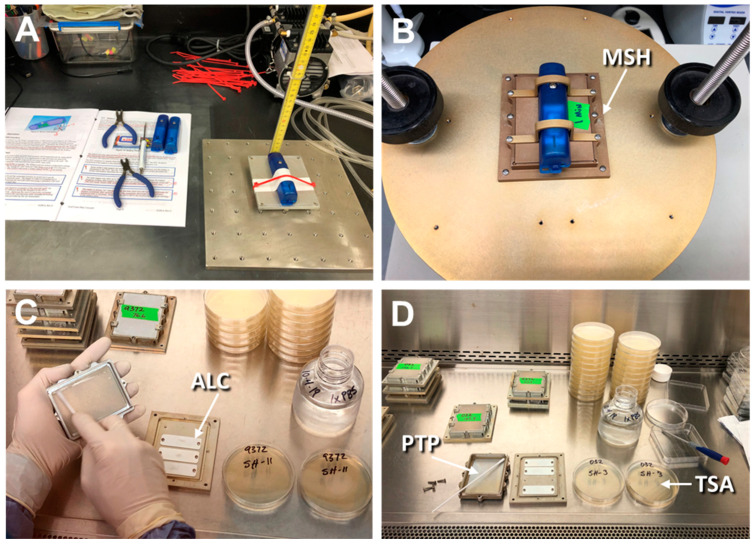
Setup for shock impact *g*-force (**A**) and vibrational *g*-force (**B**) assays. The microbial sample holders (MSH) were placed in an inverted orientation for the shock impact *g*-force assays. Conversely, the MSH units were placed in either an *Up* or an inverted *Down* orientation for the vibrational *g*-force assays. All MSH units were then processed by sampling the inside of the MSH lids (**C**) to determine if *Bacillus* spp. spores were dislodged from the doped aluminum coupons (ALC) mounted on the baseplates of the MSH units. (**D**) The polyester-tipped probe (PTP) was hydrated with filter-sterilized 1× PBS buffer prior to swabbing the inside of MSH lids and then used to streak the surfaces of two tryptic soy agar (TSA) Petri dishes per assay.

**Figure 4 microorganisms-11-02421-f004:**
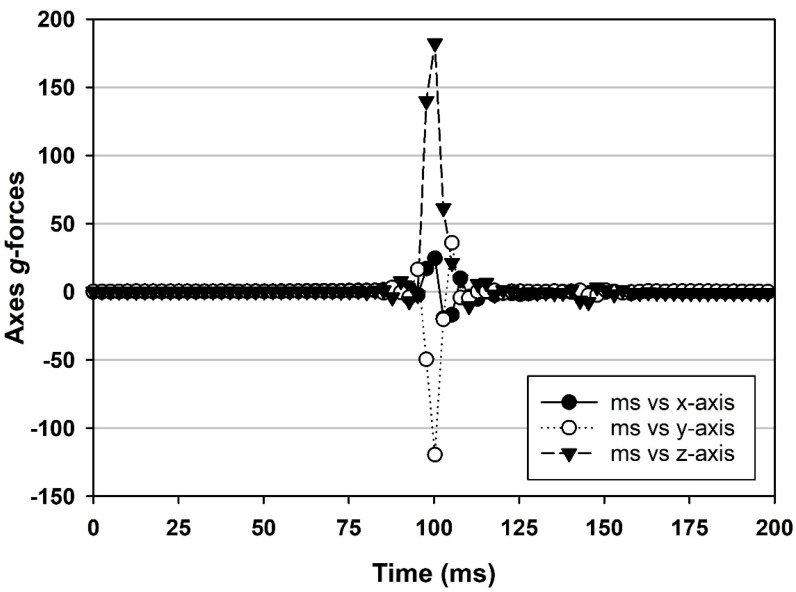
Shock-impact *g*-forces in three axes for an experiment with *Bacillus pumilus* SAFR-032 spores impacted at 182.4 *g*’s (z-axis; up/down). Results were derived from calibrated X-200A accelerometers in which the x-vector is front-to-back, the -y-vector is left-to-right, and the z-vector is up-to-down (i.e., relative to the setup in Figure 3A,B).

**Figure 5 microorganisms-11-02421-f005:**
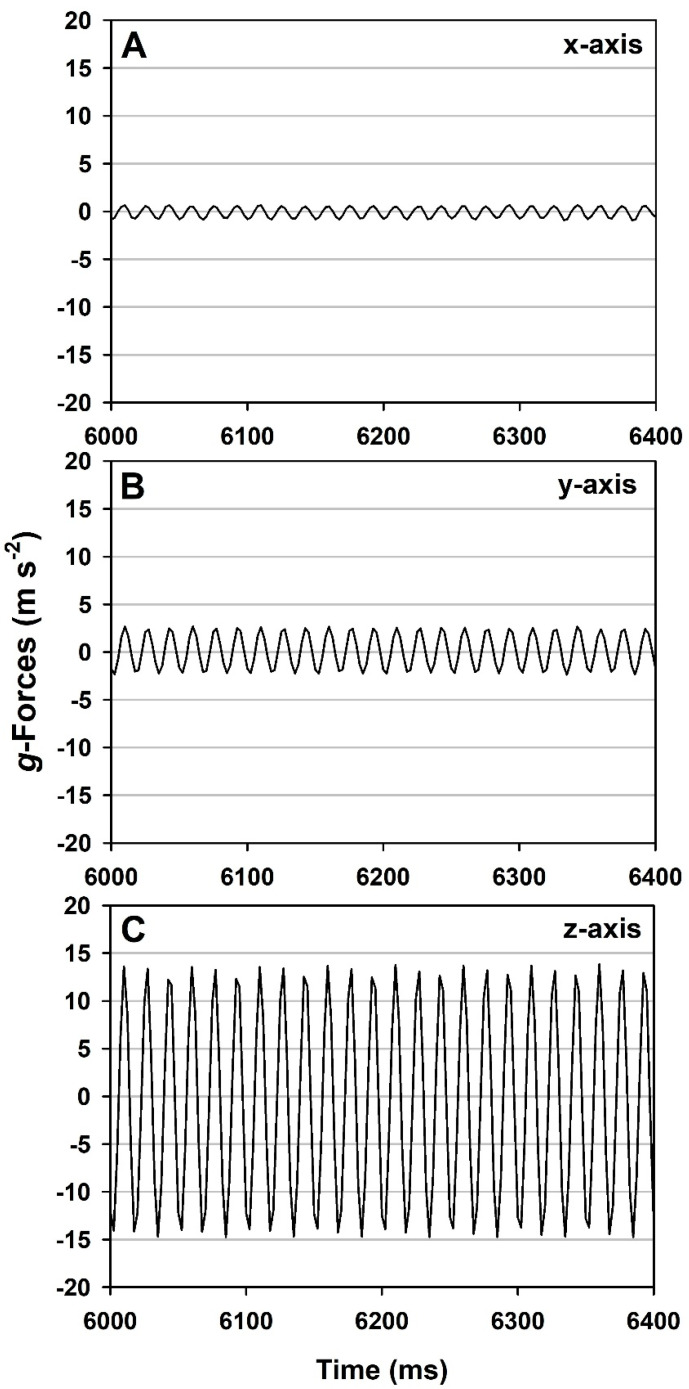
Vibrational *g*-forces in the x-, y-, and z-axes. (**A**) The vibrational forces averaged approx. 1.5 *g*’s in the x-axis (front/back). (**B**) The vibrational forces increased in the y-axis (left/right) to approx. 3–4 *g*’s. (**C**) The vibrational forces in the z-axis (up/down) were approx. 12–14 *g*’s.

**Figure 6 microorganisms-11-02421-f006:**
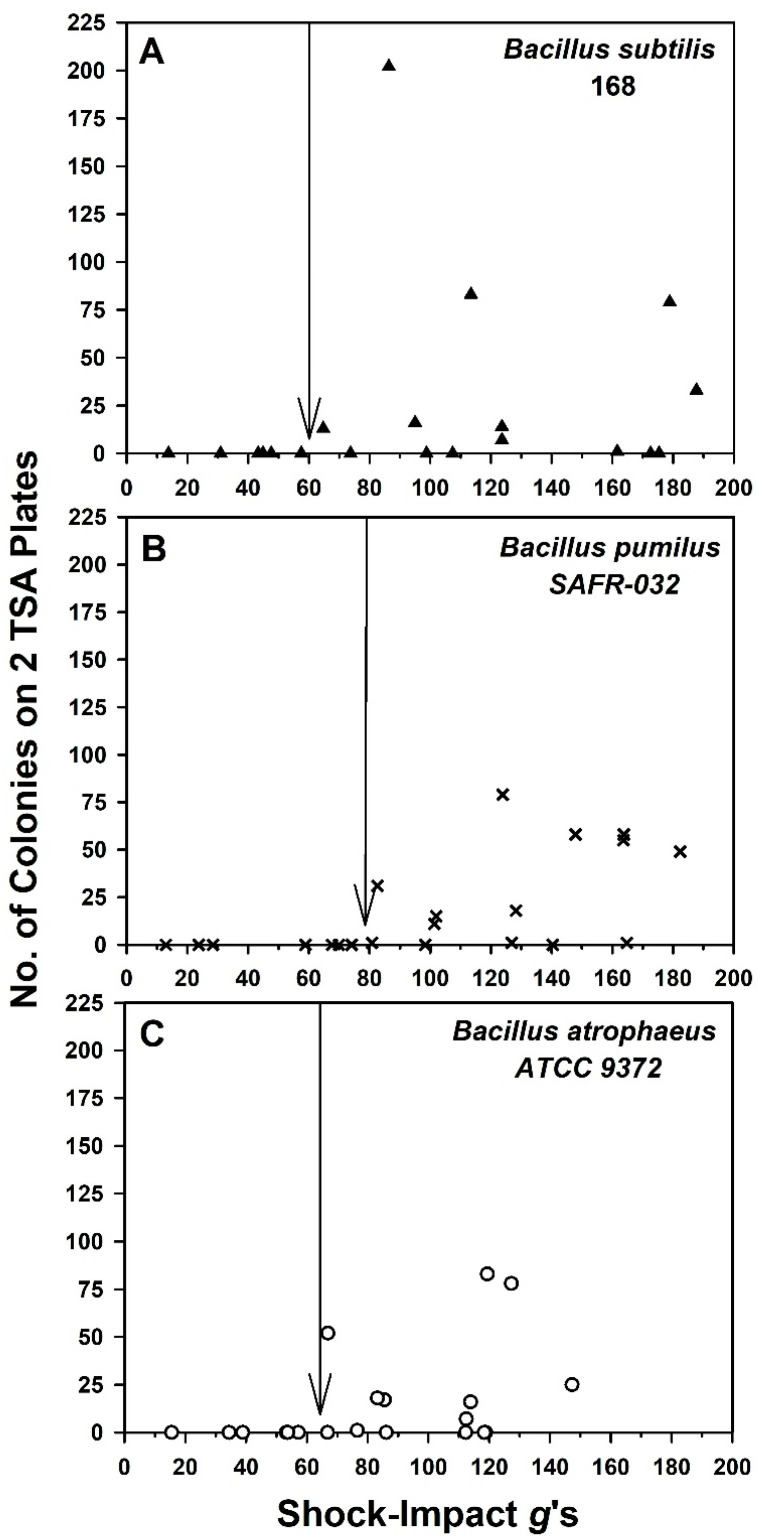
Spore removal versus shock-impact *g*-forces for the vertically accelerated microbial sample holders (MSH) containing three doped aluminum coupons per *Bacillus* spp. per MSH. Y-axes data refer to the number of cfu’s counted after two TSA plates were swabbed from the inner top surfaces of one MSH unit per assay. Symbols represent the numbers of recovered spores for each species for a single shock-impact event with one MSH unit. (**A**) Spores of *B. subtilis* 168 were dislodged at approx. 60 *g*’s. (**B**) Spores of *B. pumilus* SAFR-032 were dislodged from aluminum coupons at approx. 80 *g*’s. (**C**) Spores of *B. atrophaeus* ATCC 9372 were dislodged at approx. 65 *g*’s. (n = 20 or 21 per *Bacillus* spp. (see Appendix A)). Arrows indicate the *g*-force thresholds above which single digit to 10s or 100s of spores were dislodged for each *Bacillus* spp.

**Figure 7 microorganisms-11-02421-f007:**
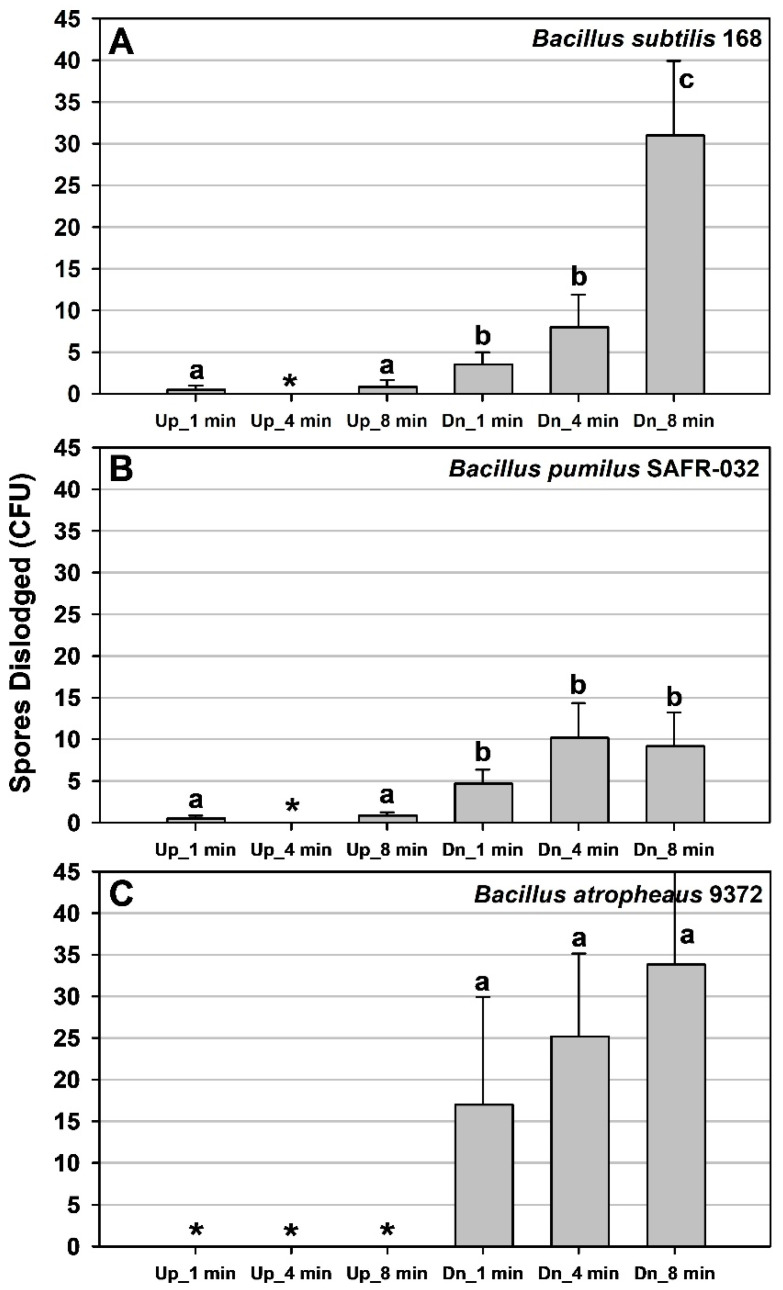
Spore removal from aluminum coupons for 10 min vibration assays. (**A**) Spores of *Bacillus subtilis* 168 were dislodged at all vibrational positions except at the *Up* + 4 min treatment (asterisk). The maximum number of spores (approx. 30 per MSH assay) were dislodged in the *Dn* + 8 min (i.e., down = *Dn*) treatment. (**B**) No spores were dislodged for *B. pumilus* SAFR-032 for the *Up* + 4 min treatments (asterisk). The maximum numbers of spores (approx. 10 per MSH assay) were dislodge in the *Dn* + 4 min treatment. (**C**) No spores were dislodged for *B. atrophaeus* ATCC 9372 for all *Up* assays (asterisks) regardless of the vibration time. The maximum numbers of spores (approx. 30 per MSH assay) were dislodged in the *Dn* + 4 min and *Dn +* 8 min treatments. Data were treated with a 0.50-power transformation to induce homogeneity of treatment variances. Transformed data were analyzed with ANOVA and a protected least squares mean separation test. Treatments for individual *Bacillus* spp. with divergent letters were significantly different at *p* ≤ 0.05 (n = 6).

## Data Availability

All raw data for Figure 6 and Figure 7 are presented as single Excel files in the Appendix A, respectively. In addition, raw data from the current study are available in a University of Florida Institutional Repository (UFIR) for A.C.S. at the link https://ufdc.ufl.edu/collections/ufir/results?page=1&q=Schuerger%2CAndrew%20 (accessed on 17 September 2023).

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
