# Peer review of "Shock-Impacts and Vibrational g-Forces Can Dislodge Bacillus spp. Spores from Spacecraft Surfaces"

_microorganisms, 2023, doi:10.3390/microorganisms11102421_

Round 1

Reviewer 1 Report

Schuerger and Borrell present here the results of impact and vibrational experiments with three different Bacillus strains. I find the manuscript in a very good and clearly structured condition. I enjoyed reading it and only wrote a few comments in the attached document. 

Author Response

Microorganisms manuscript_2598383

Title: “Shock-Impact and Vibrational g-forces can Dislodge Bacillus spp. Spores from Spacecraft Surfaces.”

Authors: Andrew C. Schuerger and Adriana V. Borrell

Reviewer #1 comments are in italics.

Responses are given after the ‘***’ in the text below.

Responses to Reviewer #1:

  • Ln 12: Is “ChemFilm” a fixed term? Otherwise explain please.  ***The term ChemFilm is used to describe a chromium oxide coating on aluminum in the spacecraft processing community.  Synonyms of ChemFilm are Iridite and Alodine.  We define the term in the methods section.
  • Ln 87-88: 100 uL suspension are forming 1.2 cm droplets? ***Yes, 100 uL of spore suspensions spread out to form a final dried ‘spot’ of spores that measure approx. 1.2 cm in diameter.  Text was added for clarity.
  • Ln 91: Open bracket missing? ***Corrected.
  • Ln 127-130: Can you say something about the effectiveness of the swab procedure? Did you measure beforehand whether all spores were absorbed by the swab and subsequently released from the swab? ***The swab protocol was a ‘semi-quantitative’ protocol to capture spores from the inside of the Microbial Sample Holder tops after reach assay.  We are confident that not all spores were recovered, and that the swab probably retained some spores on the polyester matrix during the TSA swabbing step.  But the protocol was a simple way to determine if spores were dislodged.  Text was added to describe the caveats of the using the swaps.
  • Figure 6: The description/meaning of the arrow is missing from the image description. ***Text was added to the figure legend to define the meaning of the arrows.

To All Reviewers:

  • All changes in the text are highlighted in yellow.
  • At the end of the Introduction, we reworded the last sentence to better capture the meaning of dynamic shock compressional g-forces described by citation [11]. We simplify this phrase further by using the term ‘shock impacts’ to describe the assays described for Fig. 6.
  • We also added a sentence at the end of the Introduction on the precise definition of the term ‘threshold’ used in the current text. The additional text was inserted based on feedback from colleagues on potential confusion that others might have that use the term ‘threshold velocity’.  The latter term is introduced in the Discussion. 
  • During the revision process, we had to add brief text in the Methods section to clarify the order of the figures.
  • We highlight in bold letters all occurrences of specific figures to assist in keeping them organized. The bold type can be dropped in the final published version of the paper.
  • During the revision process we noticed that there was an error in the statistics section in which we described two different approaches for the same data set presented in Fig. 7. The error has been corrected by deleting the incorrect text.  We also clarified the manner in which we handled the data in Fig. 6. 
  • We separated out the last paragraph in the Discussion and placed it into a new Conclusions section. We needed to add text to reply to comments by Reviewer #3 and creating a final Conclusions section seemed appropriate. 
  • Two citations were added to the text [new #s 23 and 24].
  • The x-axis in Fig. 6 was corrected.

Reviewer 2 Report

Line 88, Why were the coupons stored overnight before air drying?

Lines 127-129, Was one swab used to swab 2 agar plates or were two swabs used for two plates? 

Line 255, correct spelling is Phoenix.

Lines 279-287. The numbers of colony forming units were small. The method of swabbing to collect the endospores seems more qualitative than quantitative, which may be fine for this type of analysis. I would like to see a bit of discussion on this. I can imagine other methods that might be more quantitative. 

Figure 6. You have to refer to the text to realize what “No. of Colonies on 2 TSA plates” means. A figure should stand alone and be easy to understand without going to the text. Also, how many coupons were used for each organism to generate this figure? 

I may have missed it, but was the number of replicates for each experiment mentioned in the Methods section? 

English is fine.

Author Response

Microorganisms manuscript_2598383

Title: “Shock-Impact and Vibrational g-forces can Dislodge Bacillus spp. Spores from Spacecraft Surfaces.”

Authors: Andrew C. Schuerger and Adriana V. Borrell

Reviewer #2 comments are in italics.

Responses are given after the ‘***’ in the text below.

Responses to Reviewer #2:

  • Line 88, Why were the coupons stored overnight before drying? Answer: ***Storage overnight before drying allowed time for the Bacillus spores to settle in the liquid droplets and directly contact the aluminum surface before drying.  The monolayers tended to be smoother and more uniform when they were stored overnight before drying.  Text was added to the Methods clarify this point.
  • Lines 127-129, Was one swab used to swab 2 agar plates or were two swabs used for two plates? Answer: ***One swab was used to transfer spores to two TSA plates.  If we would have swabbed 3, 4, etc. TSA plates we might have recovered slightly higher numbers.  The assays were semi-quantitative.  But there was a dimensioning return on the investment of swabbing additional plates, so we kept the assays at 1 swab for 2 TSA plates.  Text was added to the Methods to help clarify this point.
  • Line 255, correct spelling of Phoenix? ***Done.
  • Lines 279-287, The numbers of colony forming units were small. The method of swabbing to collect the endospores seems more qualitative than quantitative….etc. ***The TSA assays were never designed to collect 100% of dislodged spores.  The protocol was a semi-quantitative method to characterize when any spores could be recovered from the MSH lids.  Text was added to clarify the assays qualitative versus quantitative methodology. 
  • Figure 6. You have to refer to the text to realize what “No of Colonies on 2 TSA plates” means. ***We have added text to the Figure legend to identify what this term means.
  • I may have missed it, but was the number of replicates given for each experiment mentioned in the Methods section? ***Yes, replicate numbers are given in the legends for Figs. 6 and 7.

To All Reviewers:

  • All changes in the text are highlighted in yellow.
  • At the end of the Introduction, we reworded the last sentence to better capture the meaning of dynamic shock compressional g-forces described by citation [11]. We simplify this phrase further by using the term ‘shock impacts’ to describe the assays described for Fig. 6.
  • We also added a sentence at the end of the Introduction on the precise definition of the term ‘threshold’ used in the current text. The additional text was inserted based on feedback from colleagues on potential confusion that others might have that use the term ‘threshold velocity’.  The latter term is introduced in the Discussion. 
  • During the revision process, we had to add brief text in the Methods section to clarify the order of the figures.
  • We highlight in bold letters all occurrences of specific figures to assist in keeping them organized. The bold type can be dropped in the final published version of the paper.
  • During the revision process we noticed that there was an error in the statistics section in which we described two different approaches for the same data set presented in Fig. 7. The error has been corrected by deleting the incorrect text.  We also clarified the manner in which we handled the data in Fig. 6. 
  • We separated out the last paragraph in the Discussion and placed it into a new Conclusions section. We needed to add text to reply to comments by Reviewer #3 and creating a final Conclusions section seemed appropriate. 
  • Two citations were added to the text [new #s 23 and 24].
  • The x-axis in Fig. 6 was corrected.

Reviewer 3 Report

Dear Editor and dear Authors,

Hereby, I present my comments to the manuscript “High-g Impacts and Vibrational g-Forces can Dislodge Bacillus spp. Spores from Spacecraft Surfaces” authored by Schuerger and Borrell.

Exploration missions on other planets raise the need to design protocols to avoid biological contamination, particularly if what you want to evaluate is the existence of life on said sites. Exploration protocols, to avoid contamination, can and must be improved to guarantee scientific findings, but also to minimize risk.

In this work, the authors explore the possibility that spores of model bacteria (Bacillus spp.) are transferred from one surface to another, when exposed to the gravity forces experienced by spacecrafts. The experiments simulated two scenarios: high-g shock-impacts and vibrational g-loading.

The experiments are very interesting and well designed, but I have a few questions.

1.     Did you do spore viability tests before conducting the experiments?

2.     Authors explain that a single probe was carried out per sample, but it is not clear to me if every experiment was unique or if you carried out repetitions. I assume you did because in statistical treatment there is a clue (n=6).  Could the text be more explicit?

3.     The authors mention that “The Pathfinder vehicle bounced 14 times on the Martian terrain before coming to rest.” Have you considered in your future experiments repeating the simulation but in episodes, until the total (accumulated) g value is reached?

4.     It would be interesting if your experiments were performed using other types of microorganisms. Fungi, for example, are microorganisms with an extraordinary capacity for resistance, and their resistance structures, spores, are widely distributed on the Earth's surface, hydrosphere, and atmosphere.

5.     Don’t you have some photographs of the grow of colonies? If this is the case, please add them.

Comments

1.     L.35, please delete the “;” in guidelines (i.e., lower bioburdens than required; [2,3]).”

2.     L. 71- Fig. 3. Please check the legend, is it TSA or TSB?

3.     L-90. Please check the abbreviation of hours in the text, it should be h not hrs.

4.     Petri must be written with a capital letter since it is a surname.

5.     Please change “Van Der Waals” by van der Waals.

6.     Figure 7. Please make the nomenclature used in the graphs match that described in the figure captions, to avoid possible confusion. For example: down+8 min (in the text) should be Dn_8 min (figure notation).

The document is well written. However, I detected some typos that must be corrected. 

Author Response

Microorganisms manuscript_2598383

Title: “Shock-Impact and Vibrational g-forces can Dislodge Bacillus spp. Spores from Spacecraft Surfaces.”

Authors: Andrew C. Schuerger and Adriana V. Borrell

Reviewer #3 comments are in italics.

Responses are given after the ‘***’ in the text below.

Responses to Reviewer #3:

The experiments are very interesting and well designed, but I have a few questions.

1)      Did you do spore viability tests before conducting the experiments? Answers: ***Yes.  The T = 0 numbers for coupons for all three Bacillus spp. are given in the Methods section.  The overall number of viable spores per replicate were 2 x 106 spores/coupon.  

2)      Authors explain that a single probe was carried out per sample, but it is not clear to me if every experiment was unique or if you carried out repetitions. I assume you did because in statistical treatment there is a clue (n=6).  Could the text be more explicit? *** Each datum point in Figs. 6 represents 1 MSH assay; overall the values were given as n = 20-21 for each Bacillus spp.  Furthermore, n = 6 for each bar in Fig. 7.  The numbers of replicates for both figures are in the legends.  Text was added to the Methods section for added clarity.

3)      The authors mention that “The Pathfinder vehicle bounced 14 times on the Martian terrain before coming to rest.” Have you considered in your future experiments repeating the simulation but in episodes, until the total (accumulated) g value is reached? ***This is an interesting idea, but not pursued in the current study.  It was beyond the time and budget scopes of the current study. 

4)      It would be interesting if your experiments were performed using other types of microorganisms.  Fungi, for example, are microorganisms with an extraordinary capacity for resistance, and their resistance structures, spores, are widely distributed on the Earth's surface, hydrosphere, and atmosphere. ***Agreed, but it was beyond the scope of the current paper to include other microorganisms in the study.  Furthermore, testing the same species with multiple spacecraft materials or surfaces would be an additional idea for future research.  Brief mentions of these two points are given in the Discussion.

5)      Don’t you have some photographs of the grow of colonies? If this is the case, please add them.  ***We do not have any photos of the colonies on TSA, but they appeared similar to all Bacillus spp. on TSA.  We do have images of these species on Class 1A coated aluminum coupons.  These images are presented in two previous papers (Schuerger, 2022 [citation #12] and Schuerger and Headrick, 2023 [#13]).  These papers were listed and discussed relevant to how spores might be dislodged from the monolayers of spores on the Class 1A spacecraft surfaces.  

Comments:

1)      L.35, please delete the “;” in“guidelines (i.e., lower bioburdens than required; [2,3]).”  ***Done.

2)      L. 71- Fig. 3. Please check the legend, is it TSA or TSB?  ***Done, it is TSA.

3)      L-90. Please check the abbreviation of hours in the text, it should be h not hrs.  ***Done.

4)      Petri must be written with a capital letter since it is a surname. ***Done.

5)      Please change “Van Der Waals” by van der Waals. ***Done.

6)      Figure 7. Please make the nomenclature used in the graphs match that described in the figure captions, to avoid possible confusion. For example: down+8 min (in the text) should be Dn_8 min (figure notation).  *** This was adopted in Figure 7’s legend and plots.  We spelled it out in the main text.  We were forced to abbreviate ‘down’ as ‘Dn’ in the figure axis labels because of space constraints in the SigmaPlot program. 

Comments on the Quality of English Language:  The document is well written.  However, I detected some typos that must be corrected.  ***The entire manuscript was carefully proofed for typos, syntax, and grammar.  We believe we have found and corrected a few minor edits in this regard.

To All Reviewers:

  • All changes in the text are highlighted in yellow.
  • At the end of the Introduction, we reworded the last sentence to better capture the meaning of dynamic shock compressional g-forces described by citation [11]. We simplify this phrase further by using the term ‘shock impacts’ to describe the assays described for Fig. 6.
  • We also added a sentence at the end of the Introduction on the precise definition of the term ‘threshold’ used in the current text. The additional text was inserted based on feedback from colleagues on potential confusion that others might have that use the term ‘threshold velocity’.  The latter term is introduced in the Discussion. 
  • During the revision process, we had to add brief text in the Methods section to clarify the order of the figures.
  • We highlight in bold letters all occurrences of specific figures to assist in keeping them organized. The bold type can be dropped in the final published version of the paper.
  • During the revision process we noticed that there was an error in the statistics section in which we described two different approaches for the same data set presented in Fig. 7. The error has been corrected by deleting the incorrect text.  We also clarified the manner in which we handled the data in Fig. 6. 
  • We separated out the last paragraph in the Discussion and placed it into a new Conclusions section. We needed to add text to reply to comments by Reviewer #3 and creating a final Conclusions section seemed appropriate. 
  • Two citations were added to the text [new #s 23 and 24].
  • The x-axis in Fig. 6 was corrected.
